# Impact of Introducing Hepatitis B Birth Dose Vaccines into the Infant Immunization Program in Burkina Faso: Study Protocol for a Stepped Wedge Cluster Randomized Trial (NéoVac Study)

**DOI:** 10.3390/vaccines9060583

**Published:** 2021-06-01

**Authors:** Haoua Tall, Pierrick Adam, Abdoul Salam Eric Tiendrebeogo, Jeanne Perpétue Vincent, Laura Schaeffer, Cassandre von Platen, Sandrine Fernandes-Pellerin, François Sawadogo, Alkadri Bokoum, Ghislain Bouda, Seydou Ouattara, Issa Ouédraogo, Magali Herrant, Pauline Boucheron, Appolinaire Sawadogo, Edouard Betsem, Alima Essoh, Lassané Kabore, Amariane Ouattara, Nicolas Méda, Hervé Hien, Andréa Gosset, Tamara Giles-Vernick, Sylvie Boyer, Dramane Kania, Muriel Vray, Yusuke Shimakawa

**Affiliations:** 1Epidémiologie des Maladies Evitables par la Vaccination, Agence de Médecine Préventive (AMP), Ouagadougou BP 638, Burkina Faso; htall@aamp.org (H.T.); tiendraseric@gmail.com (A.S.E.T.); fsawadogo@aamp.org (F.S.); edouardbetsem@gmail.com (E.B.); tae@aamp.org (A.E.); lassane.kabore1@yahoo.fr (L.K.); 2Unité d’Epidémiologie des Maladies Emergentes, Institut Pasteur, 75015 Paris, France; adam.pierrick@gmail.com (P.A.); jeanne.vincent@pasteur.fr (J.P.V.); laura.schaeffer@pasteur.fr (L.S.); boucheronp@fellows.iarc.fr (P.B.); muriel.vray@pasteur.fr (M.V.); 3Centre de Recherche Translationnelle, Institut Pasteur, 75015 Paris, France; cassandre.von-platen@pasteur.fr (C.v.P.); sandrine.fernandes-pellerin@pasteur.fr (S.F.-P.); 4District Sanitaire de Dafra, Ministry of Health, Bobo-Dioulasso BP 1508, Burkina Faso; alkadria1@yahoo.fr; 5District Sanitaire de Dô, Ministry of Health, Bobo-Dioulasso BP 1508, Burkina Faso; ghislainbouda@yahoo.com; 6Direction Régionale de la Santé des Hauts-Bassins, Ministry of Health, Bobo-Dioulasso BP 1508, Burkina Faso; seydououatf@gmail.com; 7Direction de la Prévention par les Vaccinations (DPV), Ministry of Health, Ouagadougou BP 7009, Burkina Faso; issayann09@yahoo.fr; 8Direction Internationale, Institut Pasteur, 75015 Paris, France; magali.herrant@pasteur.fr; 9Département de Médicine, CHU Souro Sanou, Bobo-Dioulasso BP 676, Burkina Faso; drsawadogo.appolinaire@yahoo.fr; 10Laboratoire de Virologie, Centre Muraz, Bobo-Dioulasso BP 390, Burkina Faso; koneama@yahoo.fr (A.O.); nicolas.meda@gmail.com (N.M.); hien_herve@hotmail.com (H.H.); draka3703@yahoo.fr (D.K.); 11Sciences Economiques & Sociales de La Santé & Traitement de l’Information Médicale (SESSTIM), INSERM, IRD, Aix-Marseille University, 13385 Marseille, France; andrea.gosset@inserm.fr (A.G.); sylvie.boyer@inserm.fr (S.B.); 12Unité d’Anthropologie et Ecologie de l’Emergence des Maladies, Institut Pasteur, 75015 Paris, France; tamara.giles-vernick@pasteur.fr; 13INSERM, 75013 Paris, France

**Keywords:** hepatitis B vaccine, mother-to-child transmission, birth dose vaccination, complex intervention, mixed method, stepped wedge cluster randomized trial, implementation science, sub-Saharan Africa

## Abstract

To achieve global hepatitis elimination by 2030, it is critical to prevent the mother-to-child transmission (MTCT) of hepatitis B virus (HBV). Since 2009, the WHO has recommended administering hepatitis B vaccine to all neonates within 24 h of birth to prevent MTCT. However, many countries in sub-Saharan Africa only provide hepatitis B immunization at the age of 6, 10, and 14 weeks or 8, 12, and 16 weeks using a combined vaccine. To accelerate the introduction of the hepatitis B birth dose vaccine (HepB-BD) into sub-Saharan Africa, it is critical to establish to what extent the addition of HepB-BD can further reduce HBV transmission in areas where three-dose infant vaccination has been implemented. We therefore designed a study to evaluate the impact, acceptability, and cost-effectiveness of incorporating the HepB-BD into the routine immunization program in a real-life field condition in Burkina Faso, where the hepatitis B vaccination is currently scheduled at 8-12-16 weeks. Through a multidisciplinary approach combining epidemiology, anthropology, and health economics, the Neonatal Vaccination against Hepatitis B in Africa (NéoVac) study conducts a pragmatic stepped wedge cluster randomized controlled trial in rural areas of the Hauts-Bassins Region. The study was registered in ClinicalTrials.gov (identifier: NCT04029454). A health center is designated as a cluster, and the introduction of HepB-BD will be rolled out sequentially in 24 centers. Following an initial period in which no health center administers HepB-BD, one center will be randomly allocated to incorporate HepB-BD. Then, at a regular interval, another center will be randomized to cross from the control to the intervention period, until all 24 centers integrate HepB-BD. Pregnant women attending antenatal care will be systematically invited to participate. Infants born during the control period will follow the conventional immunization schedule (8-12-16 weeks), while those born in the interventional period will receive HepB-BD in addition to the routine vaccines (0-8-12-16 weeks). The primary outcome, the proportion of hepatitis B surface antigen (HBsAg) positivity in infants aged at 9 months, will be compared between children born before and after HepB-BD introduction. The study will generate data that may assist governments and stakeholders in sub-Saharan Africa to make evidence-based decisions about whether to add HepB-BD into the national immunization programs.

## 1. Introduction

Viral hepatitis is a serious global health concern. In 2015, the World Health Organization (WHO) estimated that 1.34 million people died from viral hepatitis [1]. This is the seventh leading cause of death in the world and is ranked higher than HIV, tuberculosis, and malaria [2]. Most of these deaths occur in low-income and middle-income countries (LMIC), and more than half are attributable to chronic infection with hepatitis B virus (HBV) [1]. In 2016, the WHO adopted a strategy to globally eliminate HBV infection as a public health threat by 2030, with a goal of reducing its incidence by 90% and its mortality by 65% [3]. To reach these targets, it is crucial to eliminate the mother-to-child transmission (MTCT) of HBV, because chronic HBV infection develops more frequently when infection occurs early during childhood, especially at birth through MTCT [4,5]. Moreover, the natural history of chronic HBV infection is likely to differ depending on the mode of transmission; those who acquired chronic HBV infection through MTCT may have a higher risk of prolonged viral replication, cirrhosis and hepatocellular carcinoma (HCC) than those who acquired chronic infection through horizontal transmission [6,7,8,9].

To prevent MTCT, the WHO recommends that all neonates, regardless of maternal HBV infection, receive their first dose of the hepatitis B vaccine as soon as possible after birth, preferably within 24 h, as a monovalent formulation [10]. This hepatitis B birth dose vaccine (HepB-BD) should be followed by at least two additional doses that can be given as a monovalent or as part of a combined vaccine [10]. Most countries worldwide already integrated in their immunization programs three doses of the infant hepatitis B vaccine as a combined vaccine (pentavalent: DTP-HepB-Hib, or hexavalent formulation: DTPa-HepB-IPV-Hib) scheduled at the age of 6-10-14 weeks or 8-12-16 weeks [11]. However, access to the birth dose monovalent vaccine is still limited in many LMICs, particularly in sub-Saharan Africa, for several reasons. First, among the 47 countries in the WHO African Region, only 13 have introduced by 2020 the monovalent hepatitis B vaccination scheduled at birth [12]. This situation is partly explained by the lack of financial support; Gavi, the Vaccine Alliance, has been supporting the pentavalent vaccines but not yet the monovalent hepatitis B vaccines, because the price of the monovalent vaccine per dose is below the minimum country co-financing level ($0.20) and is thus not eligible for the Gavi [13]. Nevertheless, the Gavi will hopefully soon support strengthening the country’s vaccine delivery system to implement its timely administration within 24 h of birth [14]. Second, even a few African countries that have integrated HepB-BD in their programs have been facing difficulties in its timely administration within 24 h after birth [15] due to the high rate of child delivery at home [16,17].

More importantly, the evidence base to support the WHO’s recommendation to start immunizing immediately after birth, rather than later at 6–8 weeks of life, is limited. First, most randomized controlled trials (RCTs) evaluating the efficacy of the vaccine series starting at birth have used a control group with a placebo or without any hepatitis B vaccine administration. A meta-analysis of RCTs indicated that among infants born to HBV-infected mothers, those who received the first dose at birth are 3.5 times less likely to be infected than those who did not receive any dose of the hepatitis B vaccine (relative risk: 0.28, 95% CI: 0.20–0.40) [18]. However, this finding is not informative for a country where the hepatitis B vaccination series is currently scheduled at 6-10-14 weeks or 8-12-16 weeks, and the future introduction of HepB-BD is carefully examined; the question in such a country is to what extent the program can further prevent HBV infections by adding the monovalent HepB-BD to three doses of the combined vaccines. Second, the WHO recently conducted a systematic review of RCTs and prospective observational studies to evaluate the efficacy of the vaccine administered within 24 h of life compared to the vaccine administered after 24 h [19]. This review identified only one nonrandomized interventional study conducted by Ekra et al. in Côte d’Ivoire, which compared two vaccination schedules: 0-6-14 weeks (interventional arm with HepB-BD) and 6-10-14 weeks (control arm without HepB-BD). The study did not observe a statistically significant difference in HBsAg positivity in infants aged 9 months between the groups: 0.5% (9/1896) with HepB-BD and 0.5% (10/1900) without HepB-BD [20]. Finally, the WHO’s HepB-BD recommendation has been based on a single retrospective study in Canada [10,21]. Marion et al. studied children born to HBsAg-positive mothers, and found an increased risk of MTCT with age at the first dose (1–3, 4–7, 8–61, and ≥62 days with an odds ratio of 3.3 (95% CI: 1.3–8.2) for each unit increase in age group) [22]. The retrospective design of this study, however, poses a risk of selection bias and does not provide a satisfactory level of evidence.

Consequently, we aim to conduct a stepped wedge cluster randomized controlled trial to assess the impact of incorporating the monovalent HepB-BD into the infant immunization program on the risk of HBV MTCT under real-life field conditions in Burkina Faso, where the hepatitis B vaccination is currently scheduled at 8-12-16 weeks of life. We also assessed the acceptability of the intervention and its cost-effectiveness by adopting a multidisciplinary approach combining epidemiology, anthropology, and health economics.

## 2. Methods

### 2.1. NéoVac Research Program

Neonatal Vaccination against Hepatitis B in Africa study (NéoVac), funded by the Total Foundation, is a research program originally started in 2015 in Senegal, Burkina Faso, and Madagascar, where the hepatitis B vaccination series have been scheduled at 6-10-14 weeks (Senegal and Madagascar) or 8-12-16 weeks (Burkina Faso), without HepB-BD. The aim of the program was to develop and evaluate a strategy to implement the timely administration of HepB-BD in these countries. The program was designed in two successive steps. Step 1, the formative phase, collects epidemiological, anthropological, and economic data in each country in order to develop a locally adapted, sustainable strategy for providing timely HepB-BD within 24 h of birth. This step is followed by Step 2 to evaluate the impact of the strategy to reduce the risk of HBV infection in infants. In Burkina Faso, Step 1 was conducted in Dafra District, Hauts-Bassins Region, in 2016–2017. The epidemiological study found approximately 90% of children born at health facilities, and 86% completed three doses of the pentavalent vaccine. The anthropological study observed that infant vaccination was widely accepted in the communities, and the economic study suggested that the costs of introducing HepB-BD would be relatively low compared to the introduction costs of other vaccines [23]. These results were presented at the steering committee organized by the study investigators in Ouagadougou on 20 Nov 2017, with local healthcare workers, researchers, civil society organizations, the Ministry of Health, and the WHO. Due to a high rate of institutional delivery in the study zone, the committee considered that the administration of HepB-BD in maternity services at health centers would result in good coverage of timely HepB-BD. However, the committee questioned to what extent the addition of HepB-BD would further prevent infant HBV infection. Consequently, the overall objectives of Step 2 were defined as follows: to assess the impact of adding HepB-BD to the current hepatitis B vaccination schedule at 8-12-16 weeks of life on the risk of MTCT, and to evaluate its acceptability among healthcare workers and community members, and its cost-effectiveness.

### 2.2. Specific Objectives

The study included three work packages (epidemiology, anthropology, the health economics) funded by the Total Foundation. Each has specific objectives described below. The NéoVac program has an ancillary study called “Performance of Diagnostic Test for Hepatitis B in Africa (PREDICT-B)” funded by Gilead Sciences and Abbott. This sub-study is described at the end of Section 2.

Work package (1) Epidemiology:
-To assess the effectiveness of introducing HepB-BD on the risk of MTCT, defined as positive HBsAg in infants aged 9 months;-To assess the impact of introducing HepB-BD on the risk of MTCT in subgroups of mother–child pairs defined by the maternal HBsAg;-To assess the impact of introducing HepB-BD on the risk of MTCT in subgroups of mother–child pairs defined by the maternal hepatitis B e antigen (HBeAg) and HBV DNA levels;-To explore the dose–response relationship between the number of hepatitis B vaccine doses administered (0–4) and the risk of HBsAg positivity in infants;-To examine the association between the timeliness of the first dose of the hepatitis B vaccine and the risk of HBsAg positivity in infants;-To compare the immunological response between those who received HepB-BD and those who did not by titration of the antibody to HBsAg (anti-HBs) in infants;-To describe the coverage and timeliness of HepB-BD, pentavalent vaccines, and other vaccines recommended by the national infant immunization program in Burkina Faso;-To estimate the prevalence of HBsAg in mothers of children aged 9 months;-To estimate the prevalence of HBeAg in HBsAg-positive mothers;-To estimate the prevalence of HBV DNA levels ≥200,000 IU/mL in HBsAg-positive mothers.

Work package (2) Anthropology:
-To evaluate the acceptability of adding HepB-BD into the infant immunization program among healthcare workers and community members.

Work package (3) Health economics:
-To evaluate the public health benefits, the cost, and the cost-effectiveness of introducing HepB-BD into the infant immunization program in Burkina Faso versus the current situation.

### 2.3. Study Design

#### 2.3.1. Stepped Wedge Cluster Randomized Trial

The current protocol was developed in accordance with the CONSORT 2010 extension to cluster randomized trials [24] and the modifications specific to the stepped wedge design proposed by Hemming et al. [25]. The protocol was registered with ClinicalTrials.gov (identifier: NCT04029454), and it was reported according to the SPIRIT 2013 Statement [26]. In this multicenter stepped wedge cluster randomized controlled trial, the cluster unit is a health center (Centre de Santé et de Promotion Sociale: CSPS), a primary healthcare facility that pregnant women routinely visit for antenatal care, child delivery, and child immunization in Burkina Faso. The introduction of HepB-BD will be rolled out sequentially in 24 CSPSs in the Hauts-Bassins Region. Following an initial period in which no CSPS administers HepB-BD, one CSPS will be randomly allocated to incorporate HepB-BD into the infant immunization schedule. Then, at a regular interval of 3 weeks, another CSPS will be randomized to cross from the control period to the intervention period. This process will be repeated until all 24 CSPSs have crossed over to the intervention period (Figure 1). Other vaccines recommended by the national immunization program, including three doses of the pentavalent vaccine, will be systematically provided throughout the study period as part of the routine practice. The stepped wedge design was retained by the steering committee based on the following rationales.

#### 2.3.2. Why Cluster Randomized Trial Rather Than Individually Randomized Trial?

This is a pragmatic trial to evaluate a whole service delivery intervention that is “complex”. The intervention includes not only the administration of monovalent vaccines to newborns in a timely manner, but also the vaccine supply, storage, training of healthcare workers, and communication with pregnant women/mothers. The cluster design was chosen as it is logistically more convenient to provide the intervention at the CSPS level, rather than at the individual level. This design also enables us to limit the risk of contamination (i.e., individuals in the control group receive the intervention, and vice versa) that is more likely to happen when healthcare workers in the same facility simultaneously provide two different strategies [27].

#### 2.3.3. Why Stepped Wedge Rather Than Parallel Design?

As discussed in the introduction, the evidence supporting the addition of HepB-BD to the conventional vaccination series is limited. Nevertheless, there has been an established WHO recommendation to administer HepB-BD to all newborns. The stepped wedge design has the advantage of making all CSPSs benefit from the intervention by the end of the trial, which is politically and ethically more acceptable than the parallel cluster randomized trials, in which half of clusters do not receive the intervention for the duration of the study [25]. Moreover, the sequential rollout has fewer logistical constraints, avoiding an overcommitment of human and financial resources in comparison to the instantaneous implementation.

### 2.4. Study Setting

The trial will be conducted in two districts in the Hauts-Bassins Region, called Dô and Dafra, where an estimated 548,000 and 348,000 people lived in 2017, respectively (Figure 2). In Dô, there is one district hospital (Centre Médical avec Antenne Chirurgicale: CMA) and 28 CSPSs. In Dafra, there is one district hospital and 16 CSPSs. Each CSPS has a well-demarcated area of operation, but with population sizes that differ substantially. Consequently, each CSPS has a varying number of staff, but normally includes a chief nurse, nurses, midwives, auxiliary midwives, and mobile health workers (agent itinérant de santé: AIS). The CSPS is the primary contact between communities and the healthcare system, whilst complicated cases are referred to the district hospital where there are medical doctors. At CSPS, the antenatal care, child delivery, and child immunization are commonly shared between staff members who have a different qualification.

### 2.5. Eligibility Criteria

#### 2.5.1. Clusters

CSPSs are eligible if they are located in the rural/semi-rural zone of the Dô and Dafra districts, and they are accessible from our research center (Centre Muraz, located in Bobo-Dioulasso) all-year-round, including the rainy season. We finally selected a total of 24 CSPSs: 16 from Dô and 8 from Dafra, that met the criteria above.

#### 2.5.2. Individuals

All pregnant women of any age living in the area covered by the study CSPSs, who attended the antenatal care or gave birth at one of the study CSPSs, and who gave a written informed consent will be included. Exclusion criteria include a miscarriage, abortion, stillbirth, neonatal defect incompatible with life and any maternal or child health condition incompatible with the research activities. They will be also excluded if they had participated in other clinical trials that are incompatible with this trial.

### 2.6. Interventions

#### 2.6.1. Initial Training

Before the start of the trial, all healthcare workers from the 24 study CSPSs will be invited to the Centre Muraz for a two-day training course on the epidemiology and clinical management of hepatitis B, the administration of HepB-BD, and the study protocol including how to obtain informed consent, to fill case report forms (CRF), and to obtain blood samples. Similarly, all medical doctors from the district hospitals of Dô and Dafra will be invited for a one-day training course on the clinical management of patients with chronic HBV infection and the study protocol.

#### 2.6.2. Control Period

During the control period, there will be no supply of the monovalent vaccines to the CSPS; therefore, babies born during this period in the catchment area of this CSPS will not receive HepB-BD.

#### 2.6.3. Interventional Period

Once a CSPS crosses from the control to the intervention period, the monovalent vaccines will be delivered from the district center to the CSPS and healthcare workers will start immunizing newborns. The delivery of HepB-BD will be integrated into a monthly supply chain for the routine infant vaccines. Babies born in the CSPS during the intervention period will receive HepB-BD immediately after birth. Those born at home during the intervention period will receive HepB-BD at the first contact with the CSPS at any time up to the time when the next dose of hepatitis B vaccine is scheduled (i.e., 8 weeks as a pentavalent vaccine) according to the WHO recommendation [21].

#### 2.6.4. Routine Care

All infants, irrespective of the study period, will receive three doses of pentavalent vaccine at 8-12-16 weeks as scheduled in the national immunization program. Screening of pregnant women for HBV infection is recommended in the country but not yet systematically conducted. In the study zone, none of the CSPSs and none of the district hospitals provide antenatal screening for HBV or the treatment of chronic HBV infection using antiviral therapy. Hepatitis B immune globulin (HBIG) is not available in the study area.

### 2.7. Outcomes

#### 2.7.1. Primary Outcome

-The risk of MTCT, defined as a prevalence of positive HBsAg in infants aged 9 months.

#### 2.7.2. Secondary Outcomes

-The coverage and timeliness of HepB-BD, pentavalent vaccines, and other vaccines recommended by the national infant immunization program.-The prevalence of anti-HBs ≥10 IU/L (or ≥100 IU/L) in infants aged 9 months.-The prevalence of HBsAg in mothers of children aged 9 months.-The prevalence of HBeAg in HBsAg-positive mothers.-The prevalence of HBV DNA levels ≥200,000 IU/mL in HBsAg-positive mothers.-Anthropology: the acceptability of adding HepB-BD into the infant immunization program among healthcare workers and community members.-Health economics: the public health benefits, the cost, and the cost-effectiveness of introducing HepB-BD into the infant immunization program in Burkina Faso versus the current situation.

### 2.8. Participant Flow

#### 2.8.1. First Visit (V1)

The participant flow is schematically presented in Figure 3. Whenever a pregnant woman visits a study CSPS for antenatal care, she will be assessed for her eligibility and will be invited to participate in the study. Women coming directly to a CSPS for delivery without a prior antenatal care visit will be invited to participate immediately after the child delivery. Using a participant information sheet written in French (Appendix A) and an illustrated information sheet (Appendix A), healthcare workers will explain the study to all potentially eligible pregnant women in a local language. Women aged ≥18 years will be asked to sign an informed consent (Appendix A). Those aged <18 years will be asked to sign an assent, in addition to an informed consent form that will be signed by her legally authorized representative. Illiterate participants will be asked to provide a fingerprint. Once the informed consent is obtained, sociodemographic data will be collected through an interview with the women, and the case report form (CRF) for V1 will be filled (Appendix A).

#### 2.8.2. Second Visit (V2)

The second visit takes place at the first contact between a mother–infant pair and the CSPS. V2 will be at the time of child delivery for women who give birth at the CSPS. For women who give birth outside the CSPS, V2 will be the first postnatal care visit to the CSPS. At V2, HepB-BD will be administered to an infant who was born after his/her CSPS has crossed over to the intervention period. In contrast, HepB-BD will not be given to an infant born before his/her CSPS has started the intervention. Data on the index pregnancy, childbirth, child’s date of birth and sex, and postnatal care, including whether and when HepB-BD was administered, will be collected on the CRF for V2 (Appendix A).

#### 2.8.3. Third Visit (V3)

Mother–child pairs will be screened for HBsAg when the children return to the CSPS to receive the yellow fever vaccine and the first doses of the measles and rubella vaccines that are scheduled at the age of 9 months in Burkina Faso. Capillary blood will be obtained from mother–child pairs by finger-prick for a rapid diagnostic test (RDT) to detect HBsAg (DETERMINE™ HBsAg 2, Abbott, Abbott Park, IL, USA) [30,31] and DBS. The result of the HBsAg test will be immediately given to the participants on-site. Following the post-test counselling, 8 and 4 mL of blood will be drawn by venipuncture from the HBsAg-positive mother and HBsAg-positive child, respectively. The DBS from all participants and venous samples from those positive for HBsAg will be transferred to the Centre Muraz. Sera from infected mothers will be tested for alanine aminotransferase (ALT), aspartate aminotransferase (AST), HBV DNA PCR (RealTime HBV Viral Load Assay, Abbott), qHBsAg and qHBeAg (ARCHITECT, Abbott), HBeAg RDT (SD Bioline, Abbott), HBcrAg (Lumipulse, Fujirebio Europe, Gent, Belgium), and HBV-LAMP (Eiken Chemicals, Tokyo, Japan). Sera from infected children will be tested for ALT, AST, and HBV DNA. Vaccination history ascertained through the child vaccination card and HBsAg test result will be entered in the V3 CRF (Appendix A). Participants identified to carry HBsAg will be given a next appointment to receive results of the laboratory tests (Figure 4), while V3 will be the final study visit for those testing negative for HBsAg. Hepatitis B vaccinations will not be systematically offered to HBsAg-negative women, because the vast majority of adults in Burkina Faso have been in contact with HBV during childhood, resulting in the prevalence of total hepatitis B core antibodies (anti-HBc) exceeding 70–80% in adults [32,33].

#### 2.8.4. Participants Found to Carry HBsAg

Figure 4 presents an algorithm illustrating the clinical management of HBsAg-positive mothers and HBsAg-positive children. Once the results of ALT, AST, and HBV DNA become available, HBsAg-positive participants will be referred to the district hospital; they will be examined by general physicians, who have been trained by hepatologists at the university hospital of Bobo-Dioulasso (Centre Hospitalier Universitaire (CHU) de Sourou Sanou). If ALT is elevated (>40 IU/L), the women will be further referred to CHU where the cause of ALT elevation is investigated. Women having both ALT >40 IU/L and HBV DNA >2000 IU/mL will immediately start anti-HBV therapy (generic of tenofovir disoproxil fumarate) for their own health and will be followed at the CHU every 3 months for the duration of at least 2 years. HBsAg-positive women who do not start treatment will be followed at the district hospital every 6 months. The women known to be co-infected with HIV will be referred to the National HIV Program. A 12 month follow-up will be carried out for HBsAg-positive children at the CHU if their ALT is >40 IU/L, and at the district hospital if their ALT is ≤40 IU/L.

### 2.9. Power Calculation

The power calculation was made for the primary endpoint: the prevalence of positive HBsAg in infants aged 9 months. The study will be carried out in the rural areas of the Dafra health district (8 CSPSs covering approximately 46,000 inhabitants) and Dô health district (16 CSPSs covering approximately 154,000 inhabitants) (Figure 2). At each step, we will have a 3-week interval to allow sufficient time to include the number of participants required per CSPS and per period. To obtain the highest possible statistical power, one cluster will be randomized for each step. The following assumptions were made:(a)The average number of live births in a catchment area covered by each CSPS is about 22 per month.(b)Infant death or lost to follow-up occurs in 10% of children by the age of 9 months.(c)The prevalence of HBsAg in pregnant women (P_HBsAg_) is estimated at 10%, but it can vary from 5% to 15% between the clusters [34].(d)The prevalence of HBeAg in HBsAg-positive pregnant women (P_HBeAg_) is estimated at 15%, but it can vary from 10% to 20% between the clusters [35].(e)With three doses of the pentavalent vaccine (8-12-16 weeks) without timely administration of HepB-BD, the risk of MTCT from HBsAg-positive and HBeAg-positive pregnant women (R_HBeAg-pos_) is 60% [20,36], and the risk from HBsAg-positive and HBeAg-negative pregnant women (R_HBeAg-neg_) is 5% [35].(f)With the timely administration of HepB-BD followed by three doses of pentavalent vaccine, the risk of MTCT from HBsAg-positive and HBeAg-positive pregnant women (R_HBeAg-pos-BD_) is 20% [18], and the risk from HBsAg-positive and HBeAg-negative pregnant women (R_HBeAg-neg-BD_) is 0% [35,37].

Using assumptions (a) and (b), the number of children evaluated at 9 months of age in each observation period (3 weeks) in each cluster (CSPS) was estimated at:*n* = (Average number of live births/CSPS/month) × (1 − lost to follow-up) × (3/4 weeks) = 14.8

Using assumptions (c), (d), (e), and (f), the incidence of HBV infection among the infants aged 9 months in the intervention (π_1_) and the control (π_0_) was estimated at:π_0_ = P_HBsAg_ × P_HBeAg_ × R_HBeAg-pos_ + P_HBsAg_ × (1 − P_HBeAg_) × R_HBeAg-neg_ = 1.33%
π_1_ = P_HBsAg_ × P_HBeAg_ × R_HBeAg-pos-BD_ + P_HBsAg_ × (1 − P_HBeAg_) × R_HBeAg-neg-BD_ = 0.30%

The coefficient of variation (CV) of the incidence of HBV infection in infants aged 9 months between control clusters (control observation periods) should be determined by the variation in the prevalence of HBV among pregnant women across the clusters. The prevalence of HBsAg in pregnant women is estimated to vary by a maximum of ±5% (5–15%), and the prevalence of HBeAg in HBsAg-positive pregnant women to vary by up to ±5% (10–20%) between the clusters. Consequently, the incidence of HBV infection in infants aged 9 months in the control periods would vary between 0.53% and 2.40% according to the CSPS, with a maximum variation of 1.07% around the estimated average of π_0_ = 1.33% (i.e., π_0_ ± 2SD = 1.33% ± 1.07%). The CV, the ratio of the standard deviation (SD) to the mean, was then estimated to be 0.40 in this study. Using the formula for the power calculation of the stepped wedge cluster randomized controlled trial [38], the study will have an 84% power to demonstrate that the intervention can reduce the incidence of HBV infection in infants aged 9 months, with a significance level of 5%. This corresponds to the inclusion of approximately 8500 mother–child pairs in a recruitment period lasting for 75 weeks (24 steps × 3 weeks + 3 weeks for the last period). Alternative scenarios were considered by varying the number of infants evaluated at the age of 9 months per cluster per period for the same number of participating CSPSs (Figure 5). This shows that the sample size of 13 children aged at 9 months/cluster/period provides a power of 80%.

### 2.10. Assignment of Intervention

The chronological order in which a CSPS crosses over to the intervention will be randomized by a statistician at the Institut Pasteur, Paris, using the STATA^®^ version 15.0 (StataCorp, College Station, TX, USA). To minimize the imbalance between the intervention period and the control period, stratification will be performed by the districts (Dafra, *n* = 8; and Dô, *n* = 16). The list of the CSPS with its order will be kept in Paris, and will only be accessible by the study statistician. The intervention start date of each CSPS will be concealed from all the investigators, healthcare workers, and study participants, except the statistician. Two weeks prior to the date of the crossover from the control to the intervention, the statistician will inform the investigators about the next CSPS that will start the intervention. The study coordinator will then notify this CSPS of the intervention start date. This will ensure sufficient time for the delivery of the vaccines to the next CSPS and for the healthcare workers in this CSPS to be ready to start administering HepB-BD.

While a CSPS is in the control period, nobody will know when the intervention will begin in this CSPS. However, once CSPS has crossed over to the intervention, both the investigators and the healthcare workers in this CSPS will know that HepB-BD is now available and given in this CSPS. To limit the risk of selection bias during the enrolment of study participants, healthcare workers will inform pregnant women about the possibility of their child being vaccinated at birth, in the same way during the control and intervention period. However, despite such a precaution, if the pregnant woman spontaneously asks about the allocation status of the CSPS, the healthcare workers will tell her whether the intervention has already been assigned to this CSPS or not.

### 2.11. Data Management

At enrolment, a study identification number will be assigned to each woman. The anonymized data will be collected in a paper CRF at the CSPS, transferred to the research center at the Centre Muraz, and a double entry will be made in an electronic CRF built in the REDCap application [39]. The database will be developed and managed by the study statistician at the Institut Pasteur. All modifications made in the database will be registered according to the Good Clinical and Laboratory Practice (GCLP). Access to the final trial data and dissemination policy will be defined by the scientific committee of the NéoVac project. The final trial results will be presented at the local steering committee, and communicated with the local healthcare workers, civil societies, the Ministry of Health in Burkina Faso, the WHO country office, and the funders of the study.

### 2.12. Statistical Methods

Using an intention-to-treat analysis involving all mother–child pairs included in the study, the impact of introducing monovalent hepatitis B vaccines at birth on the risk of MTCT will be estimated by comparing the proportion of children positive for HBsAg at the age of 9 months between two groups: children born before the introduction of HepB-BD into their CSPS (control arm) and those born after their CSPS begins HepB-BD (intervention arm), regardless of whether or not they have received HepB-BD according to the assignment of the intervention to their CSPS. Our primary analysis will be “complete case analysis” excluding those with missing outcomes as it is unlikely that maternal HBV infection, the leading factor for the vaccine failure, is related with a loss to follow-up [40,41]. We will also conduct a sensitivity analysis with an extreme assumption that all those lost to follow-up were infected with HBV.

A per-protocol analysis will also be performed by restricting the analysis to children born before the introduction of HepB-BD into their CSPS and not having received HepB-BD, or to those born after the introduction of HepB-BD into their CSPS and who have received HepB-BD. To account for the calendar time and for the correlations between individuals within the same cluster, a logistic regression model with a random effect for clusters and a fixed effect for each step will be fitted [25]. All the analyses will be performed using STATA^®^.

### 2.13. Monitoring

Onsite monitoring will be performed by the team of the clinical research associate. They will monitor whether the research is conducted according to the ICH Good Clinical Practice (GCP) guidelines, including participants’ consents, cold chain, vaccine storage and administration, handling and transport of blood samples, and data quality control. Adverse events will be systematically collected and reported according to the national system. Site reports and weekly meetings with clinical research associates will be held to identify and solve the problems. A monitoring plan will be regularly reviewed and risk-based monitoring will be carried out.

### 2.14. Research Ethics Approval

The following committees rigorously evaluated the scientific, ethical, and regulatory aspects of the NéoVac study, including the PREDICT-B, and fully approved the conduct of the research:
-Comité de Recherche Clinique (CoRC), Institut Pasteur, Paris, France: approved on 25 October 2018.-IRB, Institut Pasteur, Paris, France: approved on 8 November 2018.-Comité d’éthique pour la recherche en santé (CERS), Ministère de la santé/Ministère de l’enseignement supérieur, de la recherche scientifique et de l’innovation, Burkina Faso: approved on 4 December 2018.-Comité d’éthique institutionnel du Centre MURAZ: approved on 4 April 2019.-Comité technique d’examen des demandes d’autorisation d’essais cliniques (CTEC): approved on 4 November 2019.-The project has also been registered in the ClinicalTrials.gov with the following ID (NCT04029454).

Any changes to the protocol will be submitted to these committees.

### 2.15. Dissemination Policy

The study findings will be disseminated by publishing in peer-reviewed journals, and they will be communicated locally at the steering committee with the Ministry of Health, national hepatitis program, civil society organizations, and professional associations, and internationally with key stakeholders, including the WHO.

### 2.16. Work Package 2: Anthropology

An exploratory qualitative study will be performed during the main trial to assess whether the HepB-BD strategy is likely to be accepted by communities and healthcare professionals and to explore potential barriers and facilitators to implementing the strategy. This study will use validated methods for qualitative analyses of interventions: semi-directive interviews, focus groups and participants’ observations [42,43,44]. The study will be conducted in the catchment areas of two CSPSs, one in each study district, during the intervention by trained Burkinabe anthropologists in collaboration with a senior anthropologist at the Institut Pasteur. Healthcare workers and pregnant women attending the CSPS for antenatal care will be invited to participate. Data will be collected on their comprehension of newborn care management, the biomedical category hepatitis B and any local analogous illness categories [45,46], and the vaccination. Data will be entered in NVivo 10.0 (QSR International, Melbourne, Australia), which will identify similarities and differences in describing and understanding those concepts. Findings will be interpreted overall in the light of the trial and the economic evaluation.

### 2.17. Work Package 3: Health Economics

We will assess the economic value (in USD per disability-adjusted life years (DALYs) averted) associated with the introduction of the HepB-BD strategy compared to the current vaccination schedule starting at 8 weeks (the reference strategy). The costing analysis will be performed from the health system perspective. Mathematical modelling will be used to estimate over the lifetime of the study population the health benefits (in terms of HBV infections averted, life-years, and DALYs) and the costs with and without the introduction of the HepB-BD. The cost-effectiveness of the HepB-BD strategy will be computed as the difference in costs between the two strategies divided by the difference in effectiveness (in terms of DALYs), and it will be compared to the country cost-effectiveness threshold to provide an indication of its cost-effectiveness. Health benefits will be estimated using a dynamic model that will be developed specially for the sub-Saharan African setting using standardized recommendations for the economic evaluation of immunization programs [47]. The model will estimate the dynamics of HBV transmission over time and the natural history of chronic HBV infection based on previous dynamic models [48,49]. The risks of MTCT estimated from the current trial with and without HepB-BD according to maternal HBsAg and HBeAg status will be used to model the perinatal transmission of HBV. The force of infection will be estimated from age-specific HBsAg prevalence. Infected children with chronic HBV infection will progress according to the nomenclature described by the European Association for the Study of the Liver (EASL) [50]. We will use data on costs already collected from a costing study in Step 1 [23] and newly collected from a costing study during the current trial (Step 2). Parameters that are not available in the trial, particularly the rates of disease progression in HBV-infected children, will be derived from the literature [8].

### 2.18. Ancillary Study: PREDICT-B

Using the samples collected through the main NéoVac study, this PREDICT-B study aims to assess the performance and cost-effectiveness of low-cost HBV markers to identify women with high viral load (HBV DNA ≥ 200,000 IU/mL). We have the following objectives:-To evaluate the performance of low-cost HBV markers to diagnose high-HBV DNA levels (≥200,000 UI/mL) quantified by real-time polymerase chain reaction (PCR) as a reference standard, in serum samples and dried blood spots (DBS) collected from HBsAg-positive mothers in Burkina Faso. The following alternative markers will be assessed:
◦Quantification of HBsAg (qHBsAg) and HBeAg (qHBeAg) using chemiluminescent immunoassay (CLIA) [51,52];◦Detection of HBeAg using rapid diagnostic test (RDT) [52,53,54,55,56,57,58,59];◦Quantification of hepatitis B core-related antigen (HBcrAg) [60,61,62];◦Semi-quantification of HBV DNA levels using hepatitis B loop-mediated isothermal amplification (HBV-LAMP) assay [63].-To evaluate the performance of the low-cost HBV markers in HBsAg-positive mothers to predict the risk of MTCT, defined as positive HBsAg in infants aged 9 months.-To compare the performance of the low-cost HBV markers to diagnose high HBV DNA levels (≥200,000 UI/mL) by the sample type (serum versus DBS).-To assess the effectiveness and cost-effectiveness of antenatal screening strategies using the low-cost HBV markers and subsequent antiviral treatment during pregnancy to prevent HBV MTCT in sub-Saharan Africa using modelling.

This ancillary study will use serum samples and DBS collected from HBsAg-positive mothers when their infants reach the age of 9 months, and it will test for the following markers: qHBeAg, HBeAg RDT, qHBsAg, HBcrAg, and HBV-LAMP. Then, we will evaluate the performance of these markers to diagnose the clinically important HBV DNA level (≥200,000 IU/mL) using real-time PCR (RealTime HBV Viral Load Assay) as a reference test. We will also evaluate the performance of maternal HBV markers to predict MTCT (defined as HBsAg positivity of their infants at the age of 9 months). Finally, we will assess the costs and cost-effectiveness of alternative antenatal HBV screening strategies in pregnant women based on the low-cost HBV markers.

Although it would be ideal to assess maternal HBV markers using samples collected during their pregnancy, we will evaluate these HBV markers at 9 months postpartum using the NéoVac study platform. The risk of change in maternal HBV status between pregnancy and 9 months postpartum is minimal for the following reasons:In Burkina Faso, the vast majority of adults have been in contact with HBV during childhood, resulting in the prevalence of total hepatitis B core antibody (anti-HBc) exceeding 70–80% in adults [32]. Therefore, primo-infection during adulthood is very rare.HBV viral replication remains stable after pregnancy in HBsAg-positive women; a large observational study in Australia did not find any significant difference in HBV DNA levels between samples collected during pregnancy and those collected 12 months after delivery [64].

The cost-effectiveness analysis will use real-life field data obtained from the NéoVac and PREDICT-B studies to assess the effectiveness and cost-effectiveness of alternative antenatal screening strategies using low-cost HBV markers, coupled with antiviral treatment during pregnancy. Using modelling, the effectiveness of each strategy will be estimated by the number of HBV infections averted in the cohort of children included in the NéoVac trial, as well as the number of life-years saved and DALYs saved. Costs will be estimated from a health system perspective in USD. A micro-costing study will be conducted in the study area to collect data on resources needed to implement the different screening strategies and their respective costs. The temporal horizon of the analysis will be the lifetime of the birth cohort. Table 1 summarizes the different “screen and treat” strategies that will be considered for the economic analysis. The model described in the previous section (Work package 3: Health Economics) will be used to conduct this analysis.

## 3. Discussion

This stepped wedge cluster randomized controlled trial is designed to answer two major public health questions to better fight hepatitis B in sub-Saharan Africa. The first question is what the additional impact of HepB-BD would be in reducing HBsAg prevalence in children in a country where three doses of the infant hepatitis B vaccine have been scheduled at 8-12-16 weeks. The trial will provide strong evidence to support the WHO recommendation of universal HepB-BD and to assist African countries to seriously consider integrating HepB-BD into their immunization programs. The second question is whether timely administration of HepB-BD would be enough to eliminate HBV MTCT, or whether additional measures, such as HBIG or peripartum antiviral prophylaxis, should be implemented in sub-Saharan Africa. It is well established that HBV-infected mothers with HBV DNA levels of ≥200,000 IU/mL have a substantial risk of transmitting HBV to their offspring despite infant immunoprophylaxis using both HepB-BD and HBIG [57]. As antiviral therapy during pregnancy is highly effective in preventing MTCT from these high-risk mothers [65,66], the WHO and other professional societies now recommend screening pregnant women for HBV and providing peripartum anti-HBV prophylaxis to those with high viraemia in addition to infant immunoprophylaxis at birth [50,67,68,69]. However, most of the evidence that guided these recommendations were obtained in Asia [57,66]; the applicability to sub-Saharan Africa is uncertain, given the difference between the continents in terms of HBV genotypes, natural history of chronic HBV infection, and current standards of care [11]. The NéoVac study will generate data that may inform African countries on whether to focus primarily on integrating and improving HepB-BD coverage, or whether antenatal HBV screening and peripartum antiviral prophylaxis should also be considered for the routine care of pregnant women.

In this study, we will assess the impact of universal HepB-BD, aiming to administer the monovalent hepatitis B vaccine to all neonates irrespective of maternal HBV sero-status. As an alternative strategy, selective HepB-BD, targeting neonates born to mothers who test positive for HBsAg, might be attractive and adapted to sub-Saharan Africa, because of the high uptake of antenatal care but low rate of institutional delivery in many African countries [70]. São Tomé and Príncipe, an island country in sub-Saharan Africa, introduced HepB-BD in 2002 as a selective strategy, but, recently, the Ministry of Health switched to the universal HepB-BD strategy. This was because an economic evaluation in this country found that, compared to the selective HepB-BD, the universal HepB-BD would result in a 19% reduction in chronic HBV infections per year at overall cost savings of 44% [71]. Using the data generated by NéoVac study, we will also investigate whether universal HepB-BD remains cost-effective compared to the selective HepB-BD in the study area.

## Figures and Tables

**Figure 1 vaccines-09-00583-f001:**
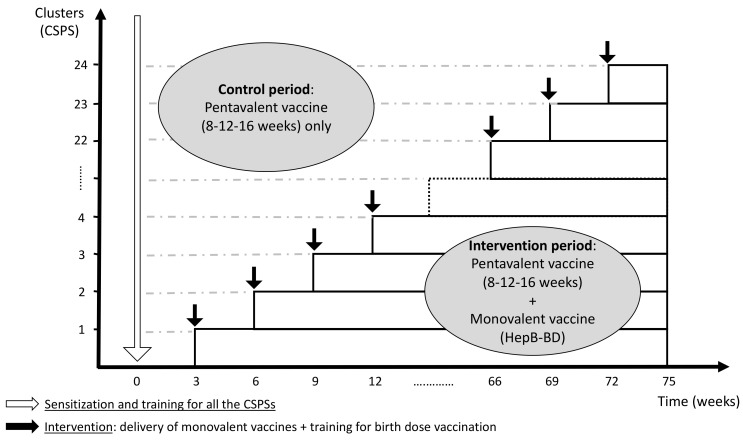
Schematic representation of the stepped wedge design (24 clusters with an interval of 3 weeks per step).

**Figure 2 vaccines-09-00583-f002:**
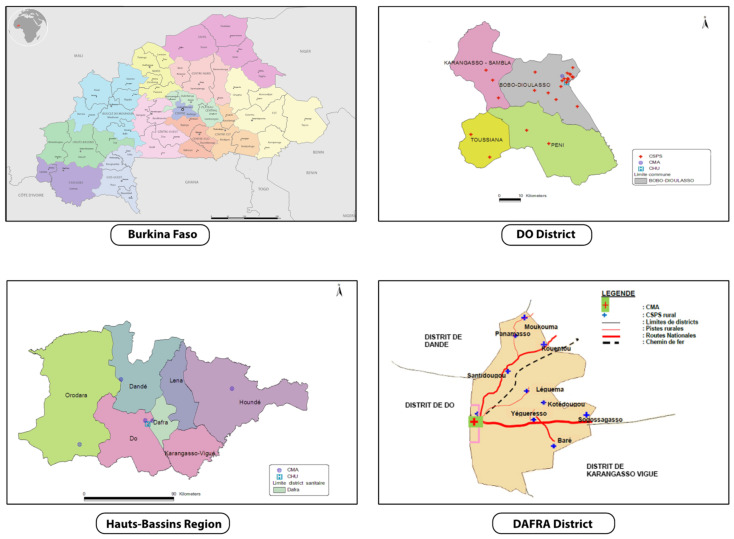
Study area (rural zones of Dô and Dafra districts in Hauts-Bassins Region). Adapted from Plan d’action sanitaire de Dô et Dafra [28,29].

**Figure 3 vaccines-09-00583-f003:**
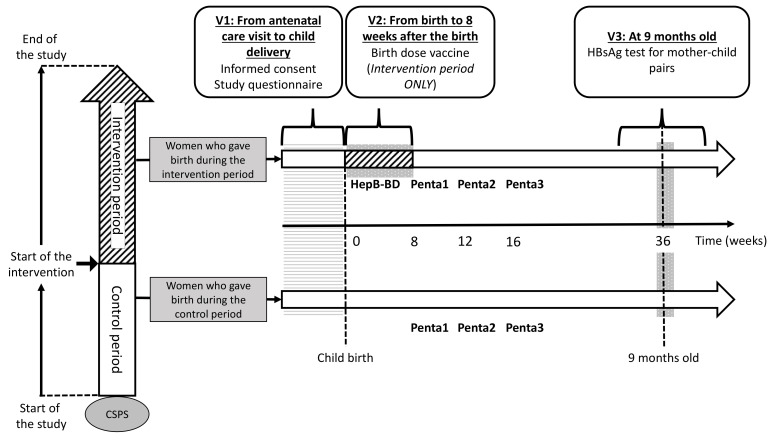
Participant timeline.

**Figure 4 vaccines-09-00583-f004:**
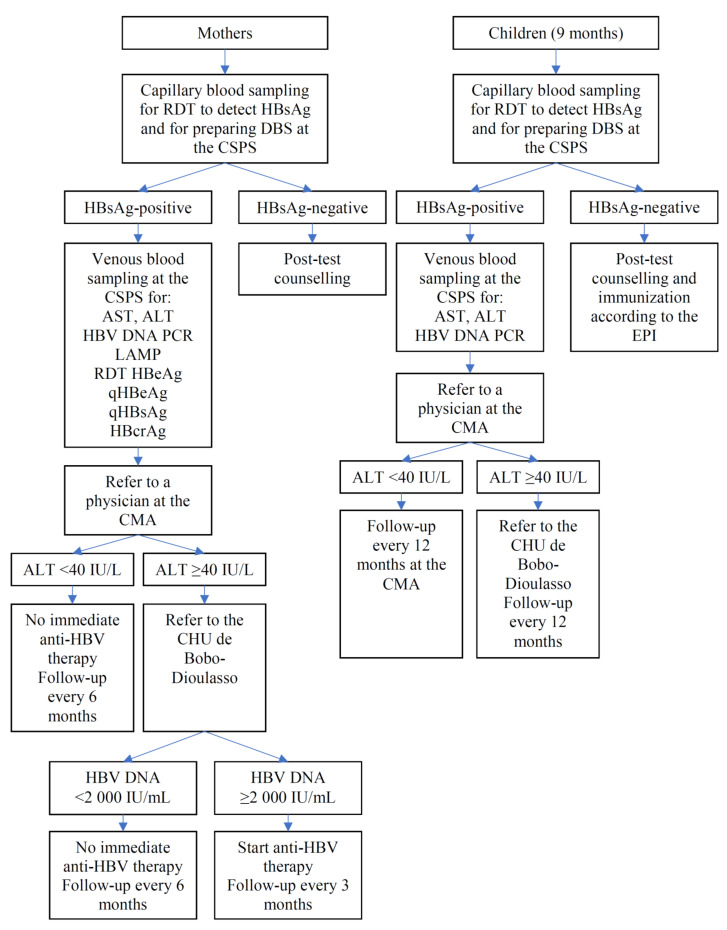
Algorithm for participants tested positive for HBsAg.

**Figure 5 vaccines-09-00583-f005:**
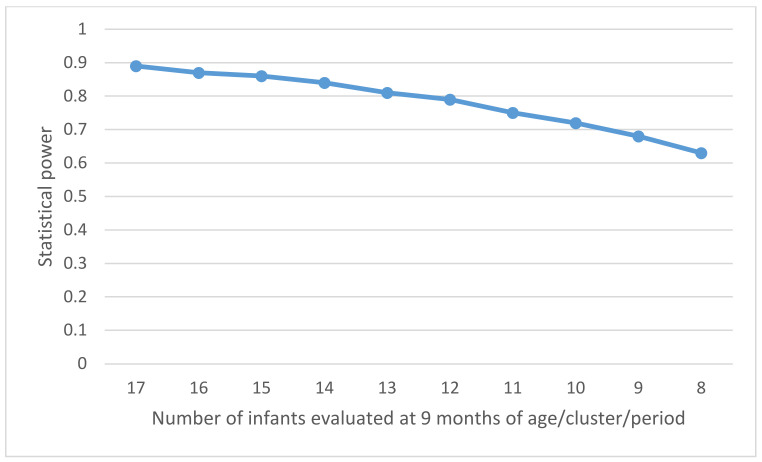
Power of the study according to the number of infants evaluated at the age of 9 months per cluster and per period.

**Table 1 vaccines-09-00583-t001:** Strategies for the prevention of mother-to-child transmission of HBV.

Strategies	Screening For Pregnant Women		Peripartum Antiviral Prophylaxis	HepB-BD	HepB3 (8-12-16 weeks)
	HBsAg RDT	Test for High Viral Replication *			
Current standard of care	-	-	-	-	Yes
Universal birth dose vaccination	-	-	-	Yes	Yes
Two-step diagnostic procedure (conventional approach in resource-rich context)	Yes	Yes (Only those tested positive for HBsAg)	Yes (Only those with high viral replication)	Yes	Yes
One-step diagnostic procedure	-	Yes (Test all women for high viral replication)	Yes (Only those with high viral replication)	Yes	Yes
Treat-all strategy	Yes	-	Yes (All HBsAg-positive women)	Yes	Yes

Abbreviations: HepB-BD, birth dose of hepatitis B vaccine; HepB3, infant vaccination with three doses of pentavalent vaccine; RDT, rapid diagnostic test. * Tests for high viral replication that we will assess are: Real-time PCR to quantify HBV DNA levels (gold standard test); qHBeAg; HBeAg RDT; qHBsAg; HBcrAg; HBV-LAMP.

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
