# Peer review of "Impact of Introducing Hepatitis B Birth Dose Vaccines into the Infant Immunization Program in Burkina Faso: Study Protocol for a Stepped Wedge Cluster Randomized Trial (NéoVac Study)"

_vaccines, 2021, doi:10.3390/vaccines9060583_

Round 1
Reviewer 1 Report
Estimated Authors,
Estimated Editors,
I've read with great interest the paper "Impact of Introducing ...". This is the report on a study protocol aimed to assess how a monovalent HBV vaccine performed in perinatal shots in high-risk children may contribute to the reduction of HBV seroprevalence.
In facts, the study is substantially well described, and I think that - once the data will have been gathered, the results may significantly improve our understanding of this topic.
In facts, I've only a couple of suggestions, and more precisely:
1) the section "Specific objectives" may be perceived as somewhat redundant with the subsequent section "outcomes". I suggest the Authors to include only one section dealing with outcomes assessed / objectives of the study.
2) some sections of the study design, i.e. subpoints 1-2, are more properly fitting discussion and conclusive remarks, as they explain the rationale behind some choices in study design/data analyses.
Author Response
Reviewer 1
I've read with great interest the paper "Impact of Introducing ...". This is the report on a study protocol aimed to assess how a monovalent HBV vaccine performed in perinatal shots in high-risk children may contribute to the reduction of HBV seroprevalence.
In facts, the study is substantially well described, and I think that - once the data will have been gathered, the results may significantly improve our understanding of this topic.
In facts, I've only a couple of suggestions, and more precisely:
- the section "Specific objectives" may be perceived as somewhat redundant with the subsequent section "outcomes". I suggest the Authors to include only one section dealing with outcomes assessed / objectives of the study.
We agree that presenting “specific objectives” first, and then “outcomes” corresponding to each of the objectives later, seems to be redundant. However, we have followed SPIRIT 2013 statement, the guidelines for good reporting of clinical trial protocol, which clearly define that the “specific objectives” and “study outcomes” need to be presented separately in a different section.
Chan AW, Tetzlaff JM, Altman DG, Laupacis A, Gøtzsche PC, Krleža-Jerić K, et al. SPIRIT 2013 statement: Defining standard protocol items for clinical trials. Ann Intern Med 2013;158:200–7.
2) some sections of the study design, i.e. subpoints 1-2, are more properly fitting discussion and conclusive remarks, as they explain the rationale behind some choices in study design/data analyses.
This is a paper reporting the protocol of the research, rather than a paper reporting the results of the research. Therefore, the rationale behind the study design is one the key elements in this work, and these need to be presented immediately after the study design.
Reviewer 2 Report
In this manuscript, Tall et al present a study protocol to elucidate the impact of administrating hepatitis B birth dose vaccines (HepB-BD) in the Hauta-Bassins region in Africa. Hepatitis B is a serious health hazard, especially in countries with low or middle income, and is the seventh leading cause of death, worldwide. Mother-to-child transmission (MTCT) is one of the prominent ways of disease transfer and can be prevented by immunizing infants on the day of birth, according to WHO. Developed nations follow this vaccination regime successfully. Sub-Saharan countries follow 6-10-14 or 8-12-16 weeks of vaccination routine due to varying reasons. Because of the absence of HepB-BD, the prevalence of hepatitis B in sub-Saharan countries appears to be higher. To facilitate the introduction of HepB-BD, it is important to establish the link between the introduction of HepB-BD and the prevention of hepatitis B in the study population. The authors here propose a systematic study wherein they will compare the proportion of hepatitis B surface antigen in infants aged 9 months, between 0-8-12-16 immunization group (HepB-BD) and the conventional group with an immunization pattern of 8-12-16 weeks. The authors hope that such a study will help create new policies regarding hepatitis vaccination in these regions/countries with low economic backgrounds.
This is a well-written study with all the aspects and prospects are discussed. I recommend acceptance in the current form.
Author Response
Reviewer 2
In this manuscript, Tall et al present a study protocol to elucidate the impact of administrating hepatitis B birth dose vaccines (HepB-BD) in the Hauts-Bassins region in Africa. Hepatitis B is a serious health hazard, especially in countries with low or middle income, and is the seventh leading cause of death, worldwide. Mother-to-child transmission (MTCT) is one of the prominent ways of disease transfer and can be prevented by immunizing infants on the day of birth, according to WHO. Developed nations follow this vaccination regime successfully. Sub-Saharan countries follow 6-10-14 or 8-12-16 weeks of vaccination routine due to varying reasons. Because of the absence of HepB-BD, the prevalence of hepatitis B in sub-Saharan countries appears to be higher. To facilitate the introduction of HepB-BD, it is important to establish the link between the introduction of HepB-BD and the prevention of hepatitis B in the study population. The authors here propose a systematic study wherein they will compare the proportion of hepatitis B surface antigen in infants aged 9 months, between 0-8-12-16 immunization group (HepB-BD) and the conventional group with an immunization pattern of 8-12-16 weeks. The authors hope that such a study will help create new policies regarding hepatitis vaccination in these regions/countries with low economic backgrounds.
This is a well-written study with all the aspects and prospects are discussed. I recommend acceptance in the current form.
Thank you very much for your comments, and appreciating our work.
Reviewer 3 Report
The present manuscript is a protocol article. The NéoVac study addresses an important public health issue, the prevention of maternal-fetal transmission of hepatitis B through vaccination at birth. The approach is intersectional and incorporates a stepped wedge cluster randomized trial, qualitative study, economic study and an ancillary study to evaluate low-cost HBV markers point of care. The publication of this complex protocol will be useful for the future presentation of the results of the Neovac study.
Minor comments:
Review the presentation of the authors' emails. Font problem
Introduction
The introduction is very clear and brings up the problematic of the study.
Methods:
The methodological choices are argued and coherent.
It is desirable to add the level of HBV viral load to the objective: “To assess the impact of introducing HepB-BD on the risk of MTCT in subgroups of mother-child pairs defined by the maternal HBsAg and hepatitis B e antigen (HBeAg)”
Among secondary outcomes “The prevalence of anti-HBs ≥10 IU/L in infants aged 9 months”, it would also be interesting to look at the rate with Ac >= 100, given their association with long term protection
Figure 4: anti-HBV therapy could be considered for all women with ALT≥ 40
Ethical considerations have been respected
Anthropological and economic methods are clearly presented and appropriate
Beyond the qualitative study, it could be interesting to study the barriers to the implementation of the dose at birth in real life
Discussion
Discuss the feasibility of implementing prenatal hepatitis B screening and targeting this intervention to positive mothers
Please discuss the role of hepatitis B vaccination for pregnant women who are not HbsAg carriers
Author Response
Reviewer 3
The present manuscript is a protocol article. The NéoVac study addresses an important public health issue, the prevention of maternal-fetal transmission of hepatitis B through vaccination at birth. The approach is intersectional and incorporates a stepped wedge cluster randomized trial, qualitative study, economic study and an ancillary study to evaluate low-cost HBV markers point of care. The publication of this complex protocol will be useful for the future presentation of the results of the Neovac study.
Minor comments:
Review the presentation of the authors' emails. Font problem
This has been done.
Introduction
The introduction is very clear and brings up the problematic of the study.
Methods:
The methodological choices are argued and coherent.
It is desirable to add the level of HBV viral load to the objective: “To assess the impact of introducing HepB-BD on the risk of MTCT in subgroups of mother-child pairs defined by the maternal HBsAg and hepatitis B e antigen (HBeAg)”
This has been added in the objectives as below:
“To assess the impact of introducing HepB-BD on the risk of MTCT in subgroups of mother-child pairs defined by the maternal HBsAg and HBV DNA levels”
Among secondary outcomes “The prevalence of anti-HBs ≥10 IU/L in infants aged 9 months”, it would also be interesting to look at the rate with Ac >= 100, given their association with long term protection
This has been added in the secondary outcomes: “The prevalence of anti-HBs ≥10 IU/L (or ≥100 IU/L) in infants aged 9 months”
Figure 4: anti-HBV therapy could be considered for all women with ALT≥ 40
The treatment indication in HBsAg-positive mothers (i.e. ALT ≥40 IU/L & HBV DNA ≥2000 IU/mL) was defined with a consultation of local hepatologists, and therefore we cannot change the criteria. In case of ALT ≥40 IU/L & HBV DNA <2000 IU/mL, a hepatologist needs to first carefully examine other causes of elevated ALT levels, including acute febrile illness such as malaria, HCV or alcohol, rather than immediate initiation of tenofovir. Moreover, the combination of ALT ≥40 IU/L & HBV DNA ≥2000 IU/mL is also used in other international guidelines as an indication for antiviral therapy.
Ethical considerations have been respected
Anthropological and economic methods are clearly presented and appropriate
Beyond the qualitative study, it could be interesting to study the barriers to the implementation of the dose at birth in real life
We have added the following in the section of anthropology: “An exploratory qualitative study will be performed during the main trial to assess whether the HepB-BD strategy is likely to be accepted by communities and healthcare professionals and to explore potential barriers and facilitators to implementing the strategy.”
Discussion
Discuss the feasibility of implementing prenatal hepatitis B screening and targeting this intervention to positive mothers
Thank you for the comment. We have added the following in the discussion.
“In this study we will assess the impact of universal HepB-BD, aiming to administer monovalent hepatitis B vaccine to all neonates irrespective of maternal HBV sero-status. As an alternative strategy, selective HepB-BD, targeting neonates born to mothers who test positive for HBsAg, might be attractive and adapted to sub-Saharan Africa, because of the high uptake of antenatal care, but low rate of institutional delivery in many African countries [57]. São Tomé and Príncipe, an island country in sub-Saharan Africa, introduced HepB-BD in 2002 as a selective strategy; but recently the Ministry of Health switched to the universal HepB-BD strategy. This was because an economic evaluation in this country found that compared to the selective HepB-BD the universal HepB-BD would result in a 19% reduction in chronic HBV infection per year at overall cost savings of 44% [58]. Using the data generated by NéoVac study, we may also investigate whether universal HepB-BD remains cost-effective compared too the selective HepB-BD.”
Please discuss the role of hepatitis B vaccination for pregnant women who are not HbsAg carriers
We have clarified in the methods section as below.
“Hepatitis B vaccination will not be systematically offered to HBsAg-negative women, because the vast majority of adults in Burkina Faso have been in contact with HBV during childhood, resulting in the prevalence of total hepatitis B core antibody (anti-HBc) exceeding 70-80% in adults. “